# Health Risks and Source Analysis of Heavy Metal Pollution from Dust in Tianshui, China

**Bo Tan** [1,2]**, Hongwei Wang** [1,2,*]**, Xinmin Wang** [3]**, Chen Ma** [1,2]**, Jing Zhou** [1,2] **and Xinyan Dai** [1,2]

[1] Key Laboratory of Oasis Ecology, Xinjiang University, Urumqi 830046, China; tanyan0829@163.com (B.T.); machen_666@163.com (C.M.); z836203@163.com (J.Z.); dxy_4150@163.com (X.D.)

[2] Key Laboratory of Smart City and Environment Modelling of Higher Education Institute, College of Resources and Environment Sciences, Xinjiang University, Urumqi 830046, China

[3] College of Resources and Environmental Engineering, Tianshui Normal University, Tianshui 741000, China; wangxm519@163.com

[*] Correspondence: wanghw@xju.edu.cn; Tel.: +86-135-7920-8666

**Abstract:** The purpose of this study is to explore the degree and spatial distribution of dust heavy metal pollution in Tianshui City, the health risks, and the sources of heavy metals. The geoaccumulation index and health risk assessment are used to study pollution levels and human health risks, and Cu, Zn, and Pb pollution are found to be serious. The total exposure of children to dust and heavy metals is $8.329 \times 10^{-3}$ mg·kg$^{-1}$·d$^{-1}$, which is 4.66 times that of adults. The effect of carcinogenic heavy metal exposure is more significant for adults than for children. The total non-carcinogenic risk quotient to children via multiple pathways is 2.1690, which is higher than that of adults. Children's Pb non-carcinogenic risk quotient is 4.79 times that of adults, and children are more sensitive than adults to the health risks of Pb pollution. The GeoDetector and Unmix 6.0 models are used for source analysis, revealing that Zn, Pb, and As pollution originate primarily from urban transportation systems, V is sourced from soil-forming parent materials, and Mn, Ni, Cu, and Co arise from mixed sources. Therefore, the treatment of heavy metal pollution in cities needs to focus more on the urban transportation system.

**Keywords:** dust heavy metals; Unmix; source; GeoDetector

## 1. Introduction

Heavy metal pollution is an important component of urban environmental pollution research. Because the urban environment is greatly affected by population concentration, developed industry and commerce, traffic congestion, and human activities, urban heavy metals are not only high in content but also present in many types. Moreover, as toxic and harmful pollutants that are difficult to degrade, heavy metals entering the soil will not only cause soil quality degradation but will also negatively affect human health through food chain accumulation, inhalation of dust, and skin contact and induce deadly diseases such as cancer [1], especially for sensitive groups such as the elderly and children [2]; thus, research on urban heavy metal pollution is necessary.

The main body of urban heavy metal pollution research includes urban surface dust [3,4], urban soil [5], green spaces [6,7], and other urban components. The research content covers content analysis, spatial variability, risk assessment [8–11], and systematic and in-depth research on the temporal and spatial properties of urban heavy metal pollution. Among this research, a study on the hazards of heavy metals in urban dust found that heavy metals in dust pose a greater risk to human health than do heavy metals in urban soil [12–15]. Among types of urban dust, road dust is an important source and sink of urban heavy metals [16], which accumulate from various sources, such as soil, air, industrial production, transportation, and coal burning [16–20]. The distribution and source of dust on urban roads vary from city to city, showing different characteristics [14,17,21].

Previous studies on heavy metal pollution in different types of cities mainly considered industrial cities [22,23] and mixed-use cities [24,25]. The characteristics of urban heavy metal pollution and possible pollution sources have been studied, but the traditional diffusion model and receptor model were mainly used in research on developed large-scale cities. Insufficient research exists on small and medium-sized cities. In addition, the analysis of pollution sources is mainly based on principal component analysis, correlation analysis, and cluster analysis. It is difficult to analyze the categories of all pollution sources, and the results are not thorough enough. It is difficult to judge the cumulative effect of natural sources and environmental particles on the content of heavy metal elements. Therefore, further in-depth research on the sources of heavy metal pollution in urban dust needs to be conducted.

Among current source analysis methods, the Unmix model, a new source analysis method, has a visual graphical interface and diagnostic tools. In contrast to the chemical mass balance (CMB) model, the Unmix model does not need to determine the source component spectrum data in advance, thus avoiding component spectrum collection and the disadvantages of high difficulty and a heavy workload [6,7]. In contrast to the PMF model, the Unmix model does not need to set the number of pollution sources and does not need to know the uncertainty of the data, which reduces the impact caused by human factors. The model obtains the results by itself according to the selected components, and the uncertainty is shown in the analysis results [26]. The Unmix model is widely used in air pollutant source analysis [27], and its accuracy and comprehensiveness have allowed it to be gradually applied to soil and sediment [28,29]. Due to the extensive sources of heavy metals in urban dust [30], traceability analysis requires the identification and analysis of multifactor sources, and geographic detectors are highly practical for such problems. Geographic detectors are based on the spatial heterogeneity of geographic phenomena, and it is assumed that if geographic factor A is controlled by geographic factor B, then B will show a spatial distribution similar to A [31,32]. Therefore, the explanatory power of influencing factors on the differentiation of heavy metals can be judged through the correlation and similarity of the influencing factors and the airborne distribution of heavy metals [33,34], and the main influencing factors can be determined. This paper uses geographic detectors in traceability analysis to reveal the influence of multiple factors.

Tianshui City is a livable construction city in Gansu Province. However, due to the development of urbanization and industrialization in recent years, the discharge of dust pollutants in the environment has increased, which directly affects the environmental quality and the health of urban residents. Compared with the sporadic distribution of super-large and large cities, such as Beijing, Shanghai, Guangzhou, and provincial capital cities, the urban system constitutes the majority of small and medium cities. Therefore, this article chose Tianshui City, an underdeveloped area in western China, as the research area. Nine heavy metal elements, Cr, Mn, Ni, Cu, Zn, As, Pb, V, and Co, were selected as the research object. The pollution degree and spatial distribution characteristics of dust heavy metals were investigated, the health risk level was assessed, and GeoDetector and the Unmix model were used to analyze the dust heavy metal pollution sources in detail. The results reveal the pollution levels and sources of heavy metals in a tourist city and provide a scientific reference for dust environmental management and heavy metal pollution control in cities.

## 2. Materials and Methods

### 2.1. Sample Collection in the Study Area

This study took the main urban areas (Maiji District and Qinzhou District) of Tianshui City as the research object. The built-up area of Qinzhou District and Maiji District is 16.4 square kilometers. The sampling points were mainly divided by the grid method, supplemented by the serpentine method, in an attempt to make the sampling points evenly distributed in the grid. Taking into account the flow of people, traffic, and other factors, a total of 51 sampling sites were set up, covering residential areas, commercial squares,

transportation hubs, hospitals, hotels, schools, and other places. In autumn 2018, sufficient surface dust samples were collected with plastic brushes and dustpans at various points near each site (approximately 10 m in radius from the center of the site), mixed, put into sample bags, and labelled for storage. A total of 51 dust samples were collected. The samples were sieved through a 180-mesh nylon sieve and then packaged in a sealed bag. The location of Tianshui City and the distribution of sampling points are shown in Figure 1. The sampling points in Qinzhou District are represented by A1, A2, A3, ... (25 in total), and those in Maiji District are represented by B1, B2, B3, ... (26 in total).

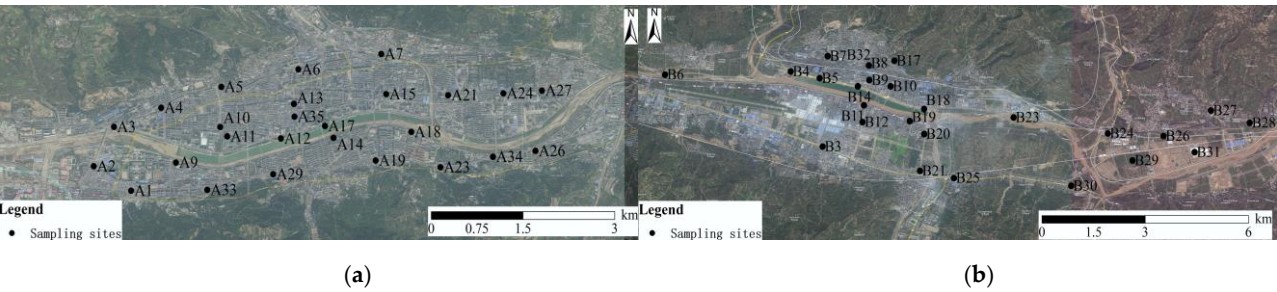

**Figure 1.** Dust sampling sites at sampling stations in Tianshui City. (**a**) Qinzhou District; (**b**) Maiji District.

*2.2. Laboratory Determination*

The dust samples were transported to the laboratory, air-dried, passed through a 0.15 mm copper sieve, and mixed evenly. Fifteen grams of sample was ground with a tungsten carbide mill (ZHM-1A) to a particle size of less than 200 mesh. The ground samples were dried at 105 °C. A total of 4 g was placed in a sample preparation mold, boric acid edging backing was added, and a 30 t semiautomatic sample press (ZHY-401A) was used to press a sample with an inner diameter of 32 mm, after which the sample was placed in a dryer until testing. A sequential wavelength dispersion-type X-ray fluorescence spectrometer (model: Axios, place of production: Almelo, Netherlands) was used to determine the element content, and the analysis software was SuperQ Version 5.0. This study selected 9 heavy metal elements (Cr, Mn, Ni, Cu, Zn, As, Pb, V, and Co) for analysis.

*2.3. Method*

2.3.1. Geoaccumulation Index

The geoaccumulation index ($I_{geo}$) was proposed by German scientists in 1969 [35]. The geoaccumulation index method takes into account the impact of human activities, environmental geochemical background values, and natural diagenesis disturbances in the evaluated system. Therefore, the geoaccumulation index method is now often used in the evaluation of soil heavy metal pollution.

The calculation formula is as follows:

$$I_{geo} = log_2(C_n/KB_n) \tag{1}$$

In the formula, $I_{geo}$ is the geoaccumulation index, $C_n$ is the heavy metal concentration in the sample, and $B_n$ is the background value of the local soil environment (due to the differences in the background values of heavy metal elements in different soil types, this article uses the heavy metal contents of limestone soil in the Longzhong area where Tianshui City is located). These element contents are used as the background values in this study [36]. K is a coefficient representing the variation in the background value caused by the difference in rocks in various places. The value of K is 1.5 in this paper [37]. The classification of the geoaccumulation index and the corresponding pollution degrees are shown in Table 1 [23,37].

**Table 1.** Geoaccumulation index classification and pollution degree.

| I$_{geo}$ Value | $\leq 0$ | 0–1 | 1–2 | 2–3 | 3–4 | 4–5 | $\geq 5$ |
|---|---|---|---|---|---|---|---|
| Contamination degree | Safe | Mild–moderate pollution | Mild–strong pollution | Moderate pollution | Strong pollution | Strong–heavy pollution | Heavy pollution |

### 2.3.2. Health Risk Assessment Method

Based on the soil health risk model proposed by the United States (US) Environmental Protection Agency (EPA) [38], the non-carcinogenic risk of 9 heavy metals at sampling points in the main urban area of Tianshui City was quantitatively estimated. Cr, Ni, and As also have carcinogenic risks. An assessment was conducted to reveal the health risks of heavy metals in dust.

The exposure calculation formula is as follows:

$$ADD_{Lng} = C \times \frac{LngR \times EF \times ED}{BW \times AT} \times 10^{-6}, \tag{2}$$

$$ADD_{Lng} = C \times \frac{LngR \times EF \times ED}{BW \times AT} \times 10^{-6}, \tag{3}$$

$$ADDdermal = C \times \frac{SA \times SL \times ABS \times EF \times ED}{BW \times AT} \times 10^{-6}, \tag{4}$$

$$LADD_{Lnh} = C \times \frac{C \times EF}{PEF \times AT} \times \left( \frac{LnhR_{child} \times ED_{child}}{BW_{child}} + \frac{LnhR_{adult} \times ED_{adult}}{BW_{adult}} \right). \tag{5}$$

where $ADD_{Lng}$, $ADD_{Lnh}$, and $ADDdermal$ are the average daily dust exposure via the human digestive system, respiratory system, and skin contact, respectively. $LADD_{Lnh}$ is the average daily dust exposure to carcinogenic heavy metals through the respiratory system. The unit of average daily dust exposure is mg/(kg·d); see Table 2 for the meaning and value of each parameter.

**Table 2.** Calculation parameters of daily average dust exposure.

| Item | Parameter | Meaning | Unit | Value | | Data Sources |
|---|---|---|---|---|---|---|
| | | | | **Child** | **Adult** | |
| Basic parameters | C | Mass fraction of heavy metals | mg/kg | Average value of study area | Average value of study area | This study |
| Exposure behavioral parameters | ED | Exposure period | a | 6 | 24 | [39] |
| | BW | Average body mass | kg | 20.3 | 58.5 | [40] |
| | EF | Exposure frequency | d/a | 180 | 180 | [41] |
| | AT (non-carcinogenic risk) | Action time | d | 365 × ED | 365 × ED | [20] |
| | AT (carcinogenic risk) | Action time | d | 365 × 70 | 365 × 70 | [20] |
| Digestive tract | IngR | Dust intake | mg/d | 250 | 150 | [42] |
| Inhalation | InhR | Air intake | m$^3$/d | 8.6 | 12.9 | [43] |
| | PEF | Particulate emission factor | m$^3$/kg | 1.36 × 10$^9$ | 1.36 × 10$^9$ | [20] |
| Skin contact | SL | Skin attachment factor | mg/cm$^2$ | 1 | 1 | [38] |
| | SA | Exposed skin area | cm$^2$/d | 1150 | 2145 | [42] |
| | ABS | Inhalation factor | — | 0.001 | 0.001 | [41] |

The non-carcinogenic and carcinogenic risks of heavy metals in dust are calculated based on the reference dose of chronic poisoning and the average lifetime daily exposure, respectively. The formulas are as follows:

$$HQ_i = ADD/RfD_i, \tag{6}$$

$$HI = \sum HQ_i, \tag{7}$$

$$Risk_i = LADD_i \times SF_i, \tag{8}$$

$$Risk_T = \sum Risk_i, \tag{9}$$

where $i$ is a specific heavy metal; $HQ_i$ is the non-carcinogenic risk quotient of heavy metal $i$ through a single route; and $ADD$ is the average daily dust. $RfD_i$ is the reference dose of a single route [38,44], which represents the maximum amount of pollutant that will not affect human health per unit time and unit body mass, mg/(kg·d).

$HI$ is the sum of the non-carcinogenic risk quotients of multiple heavy metals and multiple exposure pathways; $SF_i$ is the slope coefficient of heavy metal $I$ [38], which refers to the maximum probability of developing cancer due to human exposure to a certain dose of the heavy metal, mg/(kg·d); and $Risk$ is the carcinogenic risk quotient, indicating the possibility of cancer. If the value of $Risk$ is between $10^{-6}$ and $10^{-4}$ (that is, one cancer patient per $10^4$ to $10^6$ people), the substance is considered to have no obvious risk of cancer [20]. See Table 3 for the meaning and value of each parameter. It is generally believed that when $HQ < 1$ or $HI < 1$, the non-carcinogenic risk is small or negligible; when $HQ \geq 1$ or $HQ \geq 1$, it is considered that there is a non-carcinogenic risk.

**Table 3.** Reference dose and slope factor of different exposure pathways of heavy metals (mg·kg$^{-1}$·d$^{-1}$).

| Item | Cr | Mn | Ni | Cu | Zn | As | Pb | V | Co |
|---|---|---|---|---|---|---|---|---|---|
| RfD$_{Ing}$ | $2.86 \times 10^{-5}$ | $4.60 \times 10^{-2}$ | $2.06 \times 10^{-2}$ | $4.02 \times 10^{-2}$ | $3.00 \times 10^{-1}$ | $3.01 \times 10^{-4}$ | $3.52 \times 10^{-3}$ | $7 \times 10^{-3}$ | $2 \times 10^{-2}$ |
| RfD$_{Inh}$ | $3.00 \times 10^{-3}$ | $1.40 \times 10^{-5}$ | $2.00 \times 10^{-2}$ | $4.00 \times 10^{-2}$ | $3.00 \times 10^{-1}$ | $3.00 \times 10^{-4}$ | $3.50 \times 10^{-3}$ | $7 \times 10^{-3}$ | $1.6 \times 10^{-2}$ |
| RfD$_{dermal}$ | $6.00 \times 10^{-5}$ | $1.80 \times 10^{-3}$ | $5.40 \times 10^{-3}$ | $1.20 \times 10^{-2}$ | $6.00 \times 10^{-2}$ | $1.23 \times 10^{-4}$ | $3.25 \times 10^{-4}$ | $7 \times 10^{-3}$ | $5.06 \times 10^{-6}$ |
| SF$_{Inh}$ | 6.3 | — | 0.84 | — | — | 15.1 | — | — | — |

### 2.3.3. GeoDetector

GeoDetector is a set of statistical analysis methods that detect the spatial heterogeneity of variables, reveal the correlation between variables, and reflect the driving force behind them [31,45]. In recent years, it has also been used to study the driving factors of the spatial variability of soil heavy metals [46,47]. The GeoDetector model is composed of four submodels: a factor detector, interaction detector, risk detector, and ecological detector. This article mainly uses the core module of GeoDetector, the factor detector, to quantitatively study the effect of natural and socioeconomic factors on the heavy metal loading of dust in the study area. The influence of differentiation and the calculation method of the factor detector q value are detailed in the literature [31].

### 2.3.4. Unmix

Unmix 6.0 is a multireceptor model developed by the US EPA. It aims to solve the general mixing problem; that is, the data are a linear combination of contributions from a number of unknown sources, and the contributions of these unknown components to each sample are unknown. In this model, the singular value decomposition (SVD) method is used to reduce the dimensionality of the data space to determine the number of sources, the source composition, and the contribution rate of each source to each sample [48]. The model formulas are detailed in the literature [48].

### 2.4. Other Data Sources

Considering that dust heavy metals mainly come from the soil-forming parent material and human activities, the source factors of heavy metals in this study mainly include

the following categories: anthropogenic activity factors (daily average bus traffic ($X_1$), population density ($X_2$), distance from industrial and mining enterprises ($X_3$), road density ($X_4$), machine repair facility density ($X_5$) and natural environmental factors (distance from the river ($X_6$)), altitude ($X_7$), slope ($X_8$), $Fe_2O_3$ ($X_9$), MgO ($X_{10}$), pH ($X_{11}$), and soil texture ($X_{12}$)). Among all factors, $X_1$ and $X_4$ are factors related to the transportation system; $X_2$, $X_3$, and $X_5$ are socioeconomic factors; $X_6$, $X_7$, and $X_8$ are geographic factors, and $X_9$, $X_{10}$, $X_{11}$, and $X_{12}$ are soil parent material factors. $X_1$ is based on a Baidu map query and actual statistical results. $X_2$ comes from the Strategic Advanced Technology Products and Linkages Development Agency (https://www.worldpop.org/project/categories?id=18, accessed on 22 June 2020). $X_3$ and $X_5$ are extracted based on the place of interest (POI) data obtained from Baidu mapping and the results of buffer and nuclear density analysis. $X_4$ is based on OpenStreetMap (www.openstreetmap.org, accessed on 8 April 2020) road data obtained by nuclear density analysis. $X_6$ is obtained by buffer analysis based on OpenStreetMap river data. $X_7$ and $X_8$ are based on 30-m-resolution digital elevation model (DEM) data extracted from the geospatial data cloud hosted on the computer network of the Chinese Academy of Sciences Information Center (http://www.gscloud.cn/, accessed on 1 March 2021). Since $X_9$ and $X_{10}$ are indicator compounds of the soil parent material [49], they are used as influencing factors, and the data are based on the monitoring results for the samples in this study. $X_{11}$ and $X_{12}$ come from the China Ecosystem Assessment and Ecological Security Database (http://www.ecosystem.csdb.cn/index.jsp, accessed on 20 October 2020).

## 3. Results

### 3.1. Pollution Status Evaluation

#### 3.1.1. Analysis of Pollution Statistics

The results of the geoaccumulation index of heavy metal elements at the sampling points of the study area are shown in Figure 2. From the location of the box-shaped distribution in the figure, there is a large difference in the heavy metal $I_{geo}$ in the main urban area of Tianshui City. From the calculation results, the order of $I_{geo}$ for each element in the dust is Zn > Pb > Cu > As > Cr > Mn > V > Ni > Co. Among these elements, Mn, V, and Co are in a non-polluting state, Ni is in a light–medium pollution state, and the rest are in a safe state. The pollution statuses of Cr and Ni are similar. Cu and As predominantly exhibit mild and mild–strong pollution, accounting for 80.7% and 53.8%, respectively, and the remaining samples are in a pollution-free state. The $I_{geo}$ values of Zn and Pb range from 1.0224–5.6786 and 0.6899–4.7120, respectively, with mild–strong pollution as the main pollution degree. Zn pollution is more serious than Pb pollution. The average $I_{geo}$ values for Cr, Mn, Ni, Cu, Zn, As, Pb, V, and Co are −0.0625, −0.7191, −0.8102, 0.5390, 2.2386, 0.1449, 2.1256, −0.8126, and −1.1417, among which Cu and As exhibit light and moderate pollution, and Zn and Pb exhibit light and strong pollution.

#### 3.1.2. Spatial Distribution of Heavy Metal Pollution

The calculated $I_{geo}$ value of each sample point was loaded into ArcGIS 10.2 and the $I_{geo}$ index was spatially interpolated and mapped through Kriging analysis and histogram equalization. The results are shown in Figure 3.

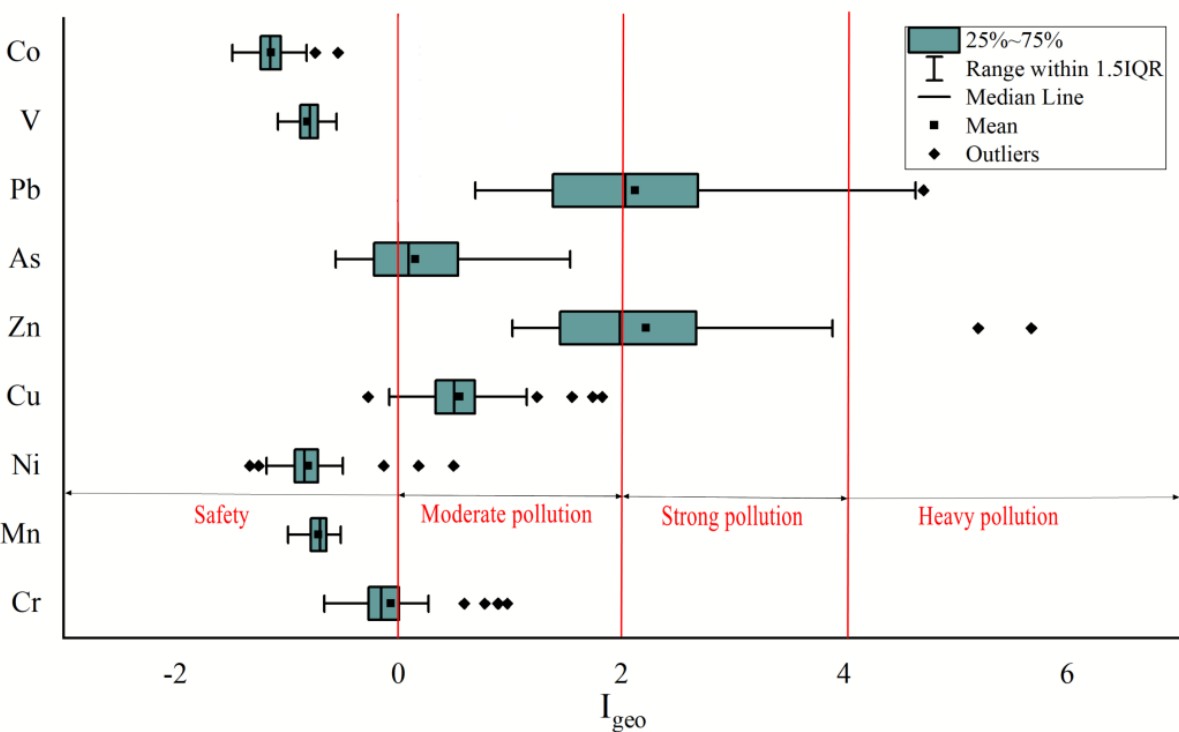

**Figure 2.** Distribution of the geoaccumulation indexes of heavy metals.

It can be seen from Figure 3 that there are significant differences in the $I_{geo}$ values of Cr, Mn, Ni, Cu, Zn, As, Pb, V, and Co in Qinzhou District (A) and Maiji District (B), and heavy metal pollution also exists in the same region. The Cr pollution in Qinzhou District is scattered, mainly concentrated in the eastern region in the areas near A21, A23, A26, A27, and A34. Cr pollution is concentrated in the Maiji District and distributed in patches. The pollution level in the central region is high, and the eastern and western regions are more polluted, with B3, B11, B12, B14, B19, and the surrounding sample sites as the main pollution points forming high pollution areas. Mn in Qinzhou District and Maiji District is characterized by strong pollution in the east and is concentrated in patches, and the western and central regions exhibit weak and strong local distribution characteristics, respectively. For Ni in Qinzhou district, A4, A10, A19, and A23 are the main high value points, and the accumulation at other sample sites is weak. The cumulative distribution of Ni in the Maiji area is similar to the distribution of Cr. The central part has a strong distribution, while the eastern and western parts have weak distributions. Cu pollution is scattered in Qinzhou District, mainly with A15, A19, A21, and A23 as strong pollution points, while Maiji District has a scattered and heavily polluted area with the western sites B9 and B5 as the main pollution locations and some polluted areas in the eastern part. Two strong pollution points are located at B24 and B27, and a weakly polluted area is located in the central part. The spatial distributions of Zn, As, and Pb pollution in Qinzhou District are similar, indicating that their sources are similar from west to east; A1, A33, A29, A12, A15, A19, A23, A26, and A27 constitute the high pollution area, while A2, A3, A4, A5, A6, and A7 form a low pollution group, and there are small differences in local samples. As a whole, the accumulation area presents a characteristic distribution of alternating strong and weak points from west to east. The pollution of As is considerably stronger than that of Zn and Pb in the central area. The $I_{geo}$ values of V and Co reach the pollution level, but the spatial distributions of these elements in the Qinzhou and Maiji districts differ, indicating that there are large differences in their sources. The V content at sampling points A18, A19, A21, A23, A33, A34, B4, B9, B10, B17, B19, and B20 is high, and the Co content is high in samples A4, A24, A26, A27, A30, B3, B5, B11, B12, and B14, indicating that the possibility of contamination is high.

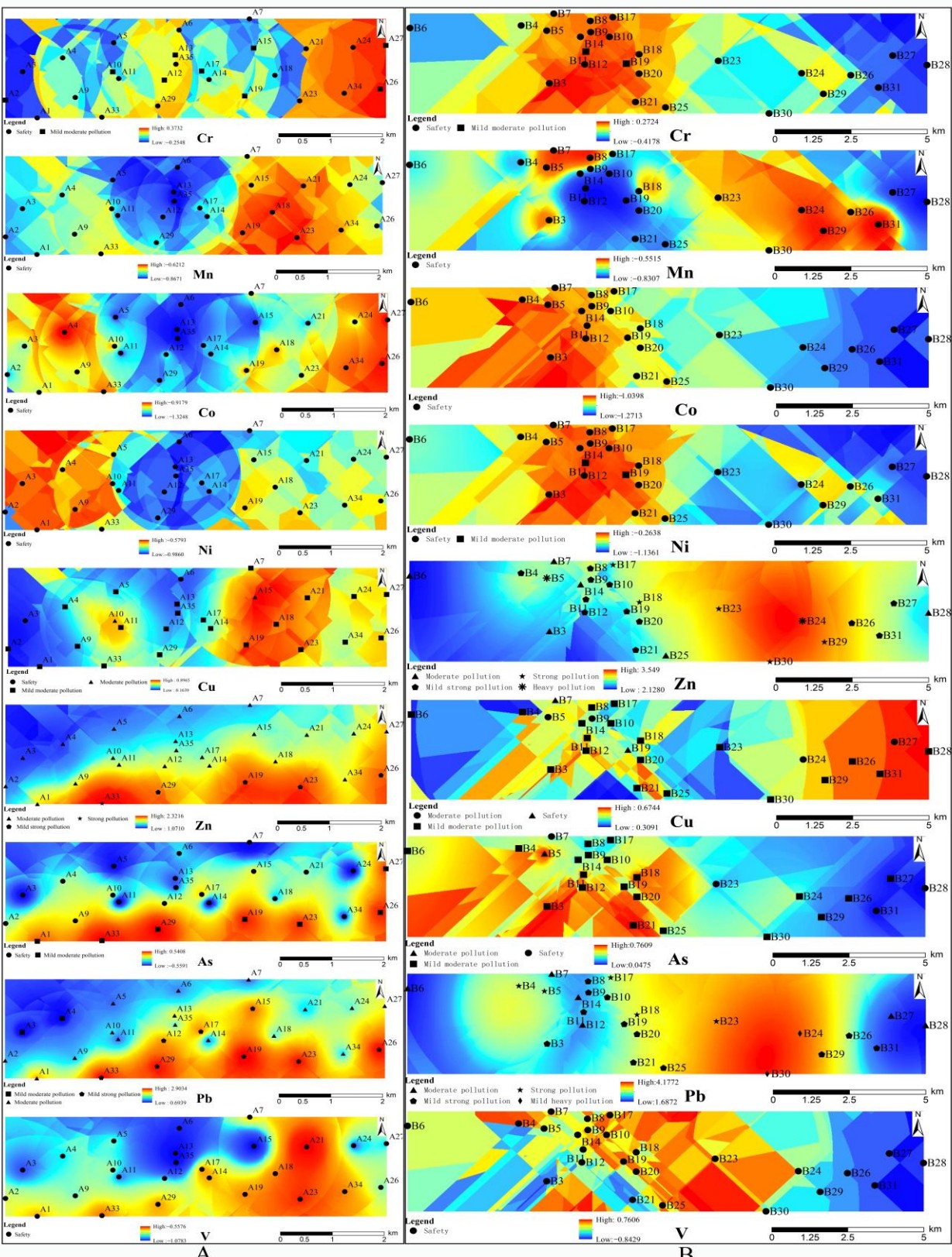

**Figure 3.** Spatial distribution of urban dust heavy metal geoaccumulation index values and pollution types in Tianshui City. (**A**) Qinzhou District; (**B**) Maiji District.

### 3.2. Health Risk Assessment

Based on the calculation of the non-carcinogenic exposure to heavy metals in dust via the digestive system, respiratory system, and skin contact and the carcinogenic exposure to some heavy metals (Table 4), the non-carcinogenic and carcinogenic risk parameters of dust and heavy metals (Table 5) were obtained. The risk value is calculated based on the average content. It should be noted that this article assumes that deposited dust is the source of inhaled dust to obtain the greatest possible health risk from inhaled deposited dust. However, the dust that naturally exists in the air is also an important source of dust inhaled by humans. In addition, this article does not consider the difference in absorbability due to the size of dust particles and uses the dust content as the inhalation amount to make a rough estimate of the risk assessment. Therefore, the results in this paper are overestimated, which means that they represent the greatest possible occurrence of health risks. The actual exposure through the respiratory system may be different from the calculated results, and the uncertainty of this assessment method needs further discussion. Although the results are exaggerated using the maximum possibility as the evaluation result, they have more value for risk warning for pollution prevention and control decision makers.

Table 4 shows that the total human exposure to heavy metals in dust follows the order As < Ni < Co < Cr < Pb < Cu < Mn < Zn < V and the exposure to non-carcinogenic heavy metals is greater than that to carcinogenic heavy metals. Among carcinogenic heavy metals, Cr has the largest exposure (65.59% of total carcinogenic heavy metal exposure), followed by Ni, and As has the least exposure (accounting for 13.99%). From the perspective of the main exposed population, the total exposure of children to heavy metals in dust is much greater than that of adults. The total exposure of children is $8.329 \times 10^{-3}$ mg·kg$^{-1}$·d$^{-1}$, which is 4.66 times that of adults ($1.786 \times 10^{-3}$ mg·kg$^{-1}$·d$^{-1}$). The exposure of children to heavy metals in dust is more significant than that of adults. Children's carcinogenic heavy metal exposure accounts for 0.903% of their total exposure, while adult carcinogenic heavy metal exposure accounts for 3.516%. The effect of carcinogenic heavy metal exposure is more significant for adults than for children; the proportions of Mn, Cu, Zn, Pb, V, and Co exposure relative to the total non-carcinogenic exposure are 38.86%, 3.82%, 39.25%, 12.20%, 4.42%, and 0.59%, respectively, for children and 38.38%, 3.77%, 38.77%, 12.04%, 4.37%, and 0.580% for adults. The effect of non-carcinogenic heavy metals on children and adults is relatively small. The exposure through the three different pathways for the different groups of people shows that the digestive system is responsible for much more exposure than skin contact and the respiratory system.

According to the risk quotients in Table 5, among children and adults, the non-carcinogenic risk quotient for exposure to all nine heavy metals via the digestive system is the highest (2.0885 for children, accounting for 96.29% of the total multipathway non-carcinogenic risk quotient, and 1.4702 for adults, accounting for 97.89%). The risk generated by the respiratory system is relatively small. The total multipathway non-carcinogenic risk quotient is 2.1690 for children and 1.5020 for adults, both of which are greater than 1, indicating that heavy metals from dust in the main urban area of Tianshui have strong non-carcinogenic effects on children and adults. The carcinogenic risk for adults is more significant than the non-carcinogenic risk, and the opposite trend is observed for children. The total non-carcinogenic risks of the nine heavy metals in descending order are Ni < Cu < Zn < Co < V<As < Mn < Pb < Cr. The worst situation is that all Cr in the dust is hexavalent. In this case, the non-carcinogenic risks of Cr to children and adults are 1.629 and 1.362, respectively, which are greater than the warning value of 1, and there is an obvious non-carcinogenic risk. The impact on human health is the strongest. However, in real conditions, the Cr is more likely to be trivalent chromium. Although excessive trivalent chromium in the human body can also cause poisoning, its toxicity is negligible compared with hexavalent chromium [50,51]. Therefore, the non-carcinogenic risk of Cr in this article is the maximum value in the theoretical sense rather than the actual risk value of dust.

**Table 4.** Dust heavy metal exposure doses at bus stations in Tianshui City ($mg \cdot kg^{-1} \cdot d^{-1}$).

| Group | Item | Cr | Mn | Ni | Cu | Zn | As | Pb | V | Co | Total |
|---|---|---|---|---|---|---|---|---|---|---|---|
| Children | $ADD_{Ing}$ | $4.65 \times 10^{-5}$ | $3.22 \times 10^{-3}$ | $1.45 \times 10^{-5}$ | $3.16 \times 10^{-4}$ | $3.25 \times 10^{-3}$ | $9.83 \times 10^{-6}$ | $1.01 \times 10^{-3}$ | $3.67 \times 10^{-4}$ | $4.87 \times 10^{-5}$ | $8.29 \times 10^{-3}$ |
| | $ADD_{Inh}$ | $1.18 \times 10^{-9}$ | $8.154 \times 10^{-8}$ | $3.66 \times 10^{-10}$ | $8.011 \times 10^{-9}$ | $8.23 \times 10^{-8}$ | $2.49 \times 10^{-10}$ | $2.56 \times 10^{-8}$ | $9.28 \times 10^{-9}$ | $1.23 \times 10^{-9}$ | $2.09 \times 10^{-7}$ |
| | $ADD_{dermal}$ | $2.14 \times 10^{-7}$ | $1.48 \times 10^{-5}$ | $6.65 \times 10^{-8}$ | $1.457 \times 10^{-6}$ | $1.50 \times 10^{-5}$ | $4.52 \times 10^{-7}$ | $4.65 \times 10^{-6}$ | $1.69 \times 10^{-6}$ | $2.24 \times 10^{-7}$ | $3.85 \times 10^{-5}$ |
| Adults | $ADD_{Ing}$ | $3.87 \times 10^{-5}$ | $6.79 \times 10^{-4}$ | $1.20 \times 10^{-5}$ | $6.594 \times 10^{-5}$ | $6.78 \times 10^{-4}$ | $8.18 \times 10^{-6}$ | $2.11 \times 10^{-4}$ | $7.63 \times 10^{-5}$ | $1.01 \times 10^{-5}$ | $1.77 \times 10^{-3}$ |
| | $ADD_{Inh}$ | $2.45 \times 10^{-9}$ | $4.24 \times 10^{-8}$ | $7.61 \times 10^{-10}$ | $4.170 \times 10^{-9}$ | $4.28 \times 10^{-8}$ | $5.18 \times 10^{-10}$ | $1.33 \times 10^{-8}$ | $4.83 \times 10^{-9}$ | $6.41 \times 10^{-10}$ | $1.12 \times 10^{-7}$ |
| | $ADD_{dermal}$ | $5.53 \times 10^{-7}$ | $9.60 \times 10^{-6}$ | $1.72 \times 10^{-7}$ | $9.43 \times 10^{-7}$ | $3.32 \times 10^{-6}$ | $1.17 \times 10^{-7}$ | $1.03 \times 10^{-6}$ | $3.74 \times 10^{-7}$ | $4.97 \times 10^{-8}$ | $1.61 \times 10^{-5}$ |
| | $LADD_{Inh}$ | $3.62 \times 10^{-9}$ | — | $1.12 \times 10^{-9}$ | — | — | $7.66 \times 10^{-10}$ | — | — | — | $6.04 \times 10^{-8}$ |

**Table 5.** Health risks of heavy metals at bus stations in Tianshui City.

| Research Object | Item | Cr | Mn | Ni | Cu | Zn | As | Pb | V | Co | HI |
|---|---|---|---|---|---|---|---|---|---|---|---|
| Children | $HQ_{Ing}$ | 1.62 | $7.00 \times 10^{-2}$ | $7.02 \times 10^{-4}$ | $7.88 \times 10^{-3}$ | $1.08 \times 10^{-2}$ | $3.26 \times 10^{-2}$ | $2.87 \times 10^{-1}$ | $5.24 \times 10^{-2}$ | $2.44 \times 10^{-3}$ | 2.09 |
| | $HQ_{Inh}$ | $3.92 \times 10^{-7}$ | $5.82 \times 10^{-3}$ | $1.83 \times 10^{-8}$ | $2.00 \times 10^{-7}$ | $2.74 \times 10^{-7}$ | $8.29 \times 10^{-7}$ | $7.31 \times 10^{-6}$ | $1.33 \times 10^{-6}$ | $7.70 \times 10^{-8}$ | $5.83 \times 10^{-3}$ |
| | $HQ_{derm}$ | $3.56 \times 10^{-3}$ | $8.23 \times 10^{-3}$ | $1.23 \times 10^{-5}$ | $1.21 \times 10^{-4}$ | $2.49 \times 10^{-4}$ | $3.68 \times 10^{-3}$ | $1.43 \times 10^{-2}$ | $2.41 \times 10^{-4}$ | $4.43 \times 10^{-2}$ | $7.47 \times 10^{-3}$ |
| | HI | 1.63 | $8.41 \times 10^{-2}$ | $7.14 \times 10^{-4}$ | $8.00 \times 10^{-3}$ | $1.11 \times 10^{-2}$ | $3.63 \times 10^{-2}$ | $3.02 \times 10^{-1}$ | $5.27 \times 10^{-2}$ | $4.67 \times 10^{-2}$ | 2.17 |
| Adults | $HQ_{Ing}$ | 1.35 | $1.46 \times 10^{-2}$ | $5.85 \times 10^{-4}$ | $1.64 \times 10^{-3}$ | $2.26 \times 10^{-3}$ | $2.72 \times 10^{-2}$ | $5.98 \times 10^{-2}$ | $1.09 \times 10^{-2}$ | $5.07 \times 10^{-4}$ | 1.470 |
| | $HQ_{Inh}$ | $8.16 \times 10^{-7}$ | $3.03 \times 10^{-3}$ | $3.81 \times 10^{-8}$ | $1.04 \times 10^{-7}$ | $1.43 \times 10^{-7}$ | $1.73 \times 10^{-6}$ | $3.80 \times 10^{-6}$ | $6.90 \times 10^{-7}$ | $4.01 \times 10^{-8}$ | $3.04 \times 10^{-3}$ |
| | $HQ_{derm}$ | $9.22 \times 10^{-3}$ | $5.33 \times 10^{-3}$ | $3.19 \times 10^{-5}$ | $7.86 \times 10^{-5}$ | $5.54 \times 10^{-5}$ | $9.51 \times 10^{-4}$ | $3.18 \times 10^{-3}$ | $5.35 \times 10^{-5}$ | $9.83 \times 10^{-4}$ | $2.87 \times 10^{-2}$ |
| | HI | 1.36 | $2.29 \times 10^{-2}$ | $6.16 \times 10^{-4}$ | $1.72 \times 10^{-3}$ | $2.31 \times 10^{-5}$ | $2.81 \times 10^{-2}$ | $6.30 \times 10^{-2}$ | $1.10 \times 10^{-2}$ | $1.03 \times 10^{-2}$ | 1.50 |
| Risk | | $2.28 \times 10^{-8}$ | — | $9.47 \times 10^{-10}$ | — | — | $1.16 \times 10^{-8}$ | — | — | — | $3.53 \times 10^{-8}$ |

This is followed by Pb, with a non-carcinogenic risk much greater than those of other elements. The non-carcinogenic risk quotient of Pb in children is 4.79 times that in adults; thus, children are more sensitive than adults to Pb pollution. Therefore, although the non-carcinogenic risk of Pb in children is lower than the warning value, there is the possibility of affecting health. Other heavy metals have a low non-carcinogenic risk quotient and have little impact on human health.

In this study, the total carcinogenic risk of the carcinogenic heavy metals Cr, Ni, and As is $5.515 \times 10^{-9}$, while the total carcinogenic risk of single heavy metals and polymetallic metals is less than $10^6$, a relatively small value.

## 4. Discussion

Determining the source of heavy metal pollution is very important for the prevention and control of heavy metal pollution. Therefore, this article first uses geographic detectors to detect the relevance of each influencing factor. Factor detection can analyze whether influencing factors have an impact on the distribution of heavy metals in dust and the size of the impact and by determining their relative importance, the main influencing factors can be screened. In this paper, factor detection was carried out on the content of heavy metals and the selected 12 factors, and the results are shown in Table 6.

**Table 6.** Explanatory power (q value) of the studied factors for eight heavy metals in soil.

| Factors | $X_1$ | $X_2$ | $X_3$ | $X_4$ | $X_5$ | $X_6$ | $X_7$ | $X_8$ | $X_9$ | $X_{10}$ | $X_{11}$ | $X_{12}$ |
|---|---|---|---|---|---|---|---|---|---|---|---|---|
| Cr | 0.3882 * | 0.1037 * | 0.0407 * | 0.097 * | 0.0603 * | 0.0505 * | 0.0005 | 0.0118 * | 0.0616 * | 0.4129 * | 0.0179 * | 0.0947 * |
| Mn | 0.2465 * | 0.1579 * | 0.0197 | 0.065 * | 0.083 | 0.0225 | 0.2366 * | 0.0711 | 0.4515 * | 0.0949 * | 0.0654 * | 0.2919 * |
| Ni | 0.2746 * | 0.0786 * | 0.083 * | 0.0836 * | 0.0465 * | 0.0257 * | 0.0063 | 0.0027 | 0.0281 * | 0.6485 * | 0.0045 ** | 0.0027 |
| Cu | 0.1327 * | 0.165 * | 0.0152 * | 0.0286 * | 0.0129 | 0.0131 | 0.0728 * | 0.0473 * | 0.0546 * | 0.0433 * | 0.0200 * | 0.2490 |
| Zn | 0.2024 * | 0.1053 * | 0.0146 * | 0.0251 * | 0.0467 * | 0.0414 * | 0.155 * | 0.0588 * | 0.1666 | 0.0188 * | 0.0841 * | 0.2850 * |
| As | 0.3751 * | 0.1663 * | 0.0178 * | 0.014 * | 0.0425 * | 0.0196 * | 0.2901 * | 0.0982 * | 0.4227 | 0.1864 * | 0.124 * | 0.5099 * |
| Pb | 0.1909 * | 0.1632 * | 0.0423 * | 0.0142 * | 0.0426 * | 0.0567 * | 0.1175 * | 0.0292 * | 0.2092 * | 0.1227 * | 0.0811 * | 0.2204 * |
| V | 0.0554 * | 0.0398 | 0.0085 | 0.0037 | 0.0267 | 0.0291 | 0.1211 * | 0.0103 | 0.4034 * | 0.1151 * | 0.1397 * | 0.2746 * |
| Co | 0.3941 * | 0.089 * | 0.0396 * | 0.0761 * | 0.0405 | 0.1038 * | 0.0006 | 0.0036 | 0.0429 * | 0.6323 * | 0.0034 | 0.0163 |

Note: the explanatory power (q value) is significant at $p < 0.05$ (**) or $p < 0.01$ (*).

The factor detection results show that most of the influencing factors passed the significance test, and the average explanatory power q was 0.1177. The distribution of heavy metals in the study area and the spatial distribution of influencing factors have macroscopic trends. The main influencing factors were extracted, and it was found that $X_{10}$, $X_1$, $X_{12}$, $X_9$, $X_2$, and $X_7$ have great explanatory power for the spatial distribution of nine heavy metals, with cumulative explanatory power values of 2.2747, 2.2598, 1.9255, 1.8406, 1.0291, and 0.9931. The content of heavy metals in the urban environment is the result of both natural background values and human activities. Therefore, from the results of factor detection analysis, the factors related to the parent material ($X_{10}$, $X_{12}$, $X_9$, and $X_7$) and socioeconomic factors ($X_1$ and $X_2$) have a significant impact on the content of heavy metals.

Ranking of the cumulative explanatory power of heavy metal elements by various influencing factors reveals that, in addition to the soil-forming parent material factor, the transportation system factor $X_1$ is the factor with the most frequent occurrences and the strongest explanatory power (cumulative explanatory power of 2.2598); furthermore, the traffic system factor $X_4$ also has strong explanatory power for the spatial differentiation of Cr, Ni, and Co content. The explanatory power of most heavy metals in terms of $X_3$ and $X_5$ is significant, but the values are all less than 0.1, at a weak level, not enough for the heavy metal content to exhibit strong space differentiation. The above results indicate that in addition to natural sources, the sources of heavy metals in the main urban area of Tianshui City are mainly from the urban transportation system and the cumulative heavy metals produced by this system are mainly due to daily operating activities such as automobile exhaust emissions and vehicle component wear.

The geographic detector effectively identified the main influencing factors of heavy metal content accumulation, but they were not enough to reveal the quantitative contribution rate of specific pollution sources to heavy metal content. Therefore, Unmix 6.0 was



used to quantitatively analyze the source composition and, combined with the geographic detector factor detection results, determine the source of heavy metals.

Unmix 6.0 software was used for source analysis. When all nine heavy metals were analyzed, the software prompted that there was no solution. After removing Cr according to the software recommendations, the model ran normally, and the results show that the system has four sources and that the model Min $R^2$ = 0.97. A total of 97% of the species variance can be explained by the model, which is greater than the minimum value required by the system (Min $R^2$ > 0.8). Min Sig/Noise = 2.09, which is greater than the minimum value required by the system (Min Sig/Noise > 2). The analytical results from these four sources are credible. The results of geographic detection in the previous section indicate that the content of heavy metals is mainly affected by natural sources (soil-forming parent material) and transportation system factors. Therefore, the soil-forming parent material and transportation system factors were used as the main analytical factors in the source analysis process, and other factors were used as references in conjunction with the source contribution data. The similarity between the spatial distributions of sources and high-load pollutants was used to determine the sources of heavy metal pollution. The spatial distribution of source contributions is shown in Figure 4.

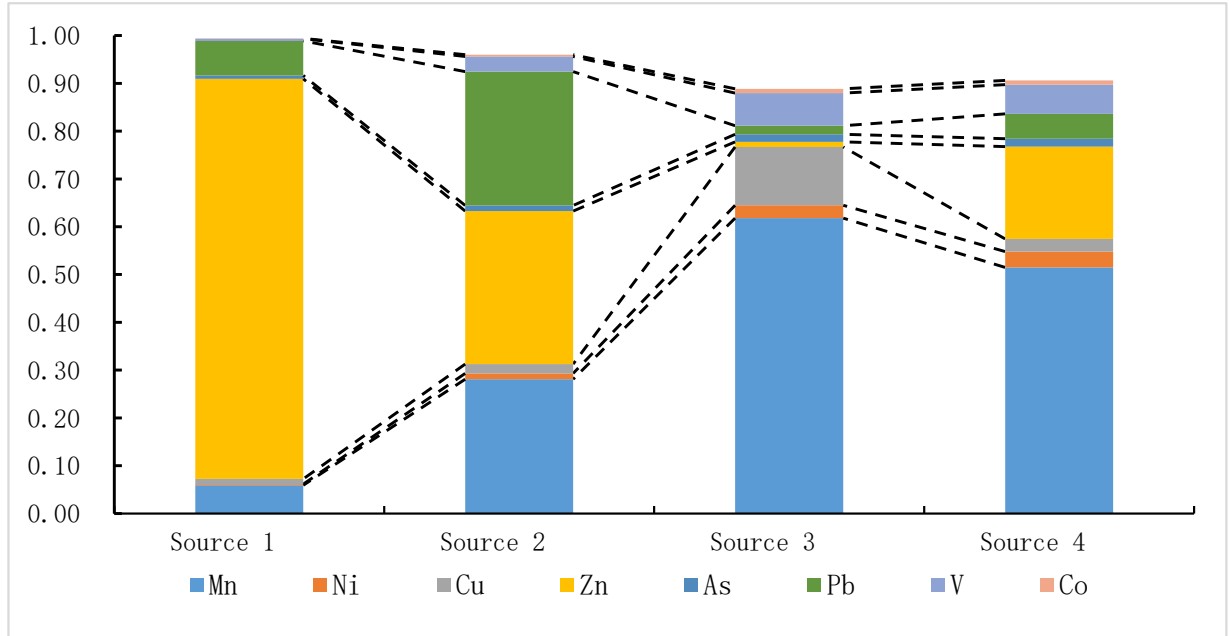

**Figure 4.** Contribution rates of different sources to heavy metal pollution in dust.

Source 1 has the highest Zn loading in the composition spectrum, reaching 83.7%. Source 1 causes Zn accumulation. Factor detection analysis shows that apart from natural sources of Zn, the daily average bus traffic has the strongest explanatory power, with the q value reaching 0.2024, indicating that operation of the transportation system has the greatest influence on the accumulation of Zn. Zn-containing castings are important materials for automobile transmission parts, engine parts, and body components in the automobile industry. Zn-containing roadside soil produced by vehicle wear and tear accumulates in surface dust, causing Zn pollution [45]. Moreover, the contribution rate of source 1 to sample points 29, 39, and 44 in Figure 5 is significantly greater than that for other sample points. These sample sites are adjacent to the main roads in Maiji District. These roads are the first-class roads in the region and bear the main passenger and freight transportation in the city. The roads have a large traffic capacity and are a section that is prone to congestion during rush hours. Passenger and freight vehicles emit large amounts of exhaust gas, mechanical friction and tire wear are serious, and Zn accumulation is serious. Studies have shown that Zn may come from roofs and gutters [52,53], which may

be related to the Zn-containing materials used in roof construction, especially galvanized materials and coatings [54]. Zn accumulates in drains along with roof runoff. However, the roofing materials used in Tianshui City are mainly expanded polystyrene boards, polymer cement waterproof coatings, and asphalt. There are fewer Zn-containing materials, and the possibility of Zn coming from roofs and drainage ditches is less likely. Therefore, source 1 represents the accumulation of heavy metals generated by the wear of automobile parts in the transportation system.

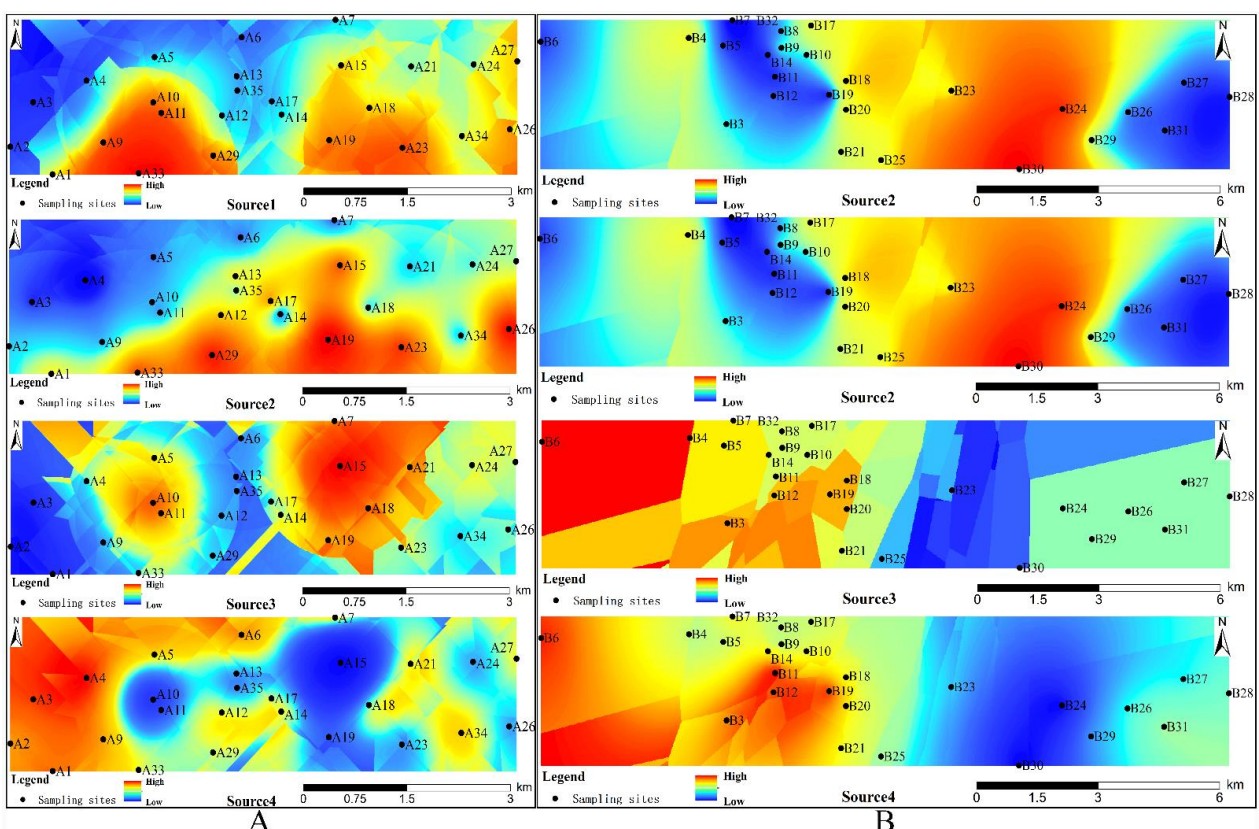

**Figure 5.** Source contribution spatial distribution. (**A**) Qinzhou district; (**B**) Maiji district.

The loading ratios of Mn, Zn, and Pb in the Source 2 composition spectrum are 28.1%, 32.0%, and 28.0%, respectively. Source 2 is the primary source of Pb pollution and has a dominant contribution to Pb content. In the transportation system, automobile exhaust is a direct source of Pb [55]. At present, the fuel used by Chinese cars is #92 or #95 gasoline and #0 diesel. Pb is the most important heavy metal in the exhaust of four vehicle types [49]. The heavy metals in exhaust gas also include Mn, Zn, Ni, As, Cr, Co, and others. Existing studies have shown that the content of Mn in exhaust gas from #92 and #95 gasoline is significantly higher than that of other heavy metals [55]. Exhaust gas is an important source of Mn in dust. Therefore, it is believed that source 2 is automobile exhaust in the transportation system.

The load of Mn in the composition spectrum of source 3 is 61.8%, the Cu load is 12.2%, and the V load is 6.8%; these values are far greater than the contribution rates of other sources to these three heavy metals, so source 3 is the dominant source of the above three elements. The Zn and Pb loadings in source 3 are both less than 2%, indicating that traffic system pollution sources account for a relatively small proportion in source 3; thus, traffic system pollution sources are excluded. The factor detection analysis results (Table 6) show that the primary factors leading to the accumulation of Mn, Cu, and V are the soil parent material factors (soil texture or $Fe_2O_3$) and that the main influencing factors of V are the soil parent material factors and other factors. Natural factors are characterized by strong

natural sources. These three elements mainly come from natural environmental conditions, and thus source 3 can be identified as a natural source.

Source 4 has the largest loading of Mn and Zn in the composition spectrum, but its contribution rate to the accumulation of Ni is the highest among the four sources, which is 3.3%. In a study of the sources of Ni in cities, it was found that Ni was mainly derived from natural sources such as soil-forming parent material [56] and industrial activities [57]. The factor detection results (Table 6) show that the content of the soil-forming parent material factor MgO and socioeconomic factors such as daily average bus traffic, road density, distance to industrial and mining enterprises, and population density have significant explanatory power for the accumulation of Ni. The cumulative explanatory power of the soil-forming parent material factor and the socioeconomic factor are 0.6485 and 0.5198, respectively. The difference in influence intensity is small. The soil parent material and urban socioeconomic activities, including industrial activities, are contributing factors.

Regarding the high contribution rate of Mn and Zn, it can be seen from the factor detection results that Mn and Zn cover natural and anthropogenic sources. When the dominant source is excluded, the cumulative effect of heavy metals produced by other mixed sources also appears. In the source 4 component spectrum, the Cu load is 2.6%, second only to that of source 3.

However, existing urban heavy metal traceability studies have shown that the sources of Cu are diverse, including factors such as man-made emissions [58], brake materials [59], construction activities [55], and atmospheric dust fall [60]. The factor detection results (Table 6) indicate that X1 and X2 have strong explanatory power for Cu accumulation. It is believed that Cu is not only from natural sources but also from urban construction and development and urban daily economic activities. Therefore, source 4 is considered to be a mixed source composed of natural and manmade sources.

The loading of Co in each source component spectrum shows that the loadings of Co in source 3 and source 4 are both 0.009, so source 3 and source 4 are the pollution sources of Co. It is generally believed that in addition to natural accumulation during soil formation, Co mainly comes from industrial production, such as alloy production, electroplating, glass manufacturing, dyeing and other industries, with a wide range of sources, so Co has certain loadings in natural sources and mixed sources.

Since Cr was excluded in the Unmix model analysis, principal component analysis was used to assess the traceability of Cr, and the results were used to verify the above traceability analysis results. Three principal components were extracted by SPSS principal component analysis, and the cumulative contribution rate of the three principal components was 78.088%. The three principal components basically reflect the variability of the nine elements studied.

It can be seen from Table 7 that the elements with higher loads in Factor 1 are Zn, Pb, and As; the elements with higher loads in Factor 2 are Cr, Ni, and Co; and the elements with higher loads in Factor 3 are Mn, V, and Co. Since heavy metals with higher factor loadings under the same principal component have the same source, according to the source types of heavy metal elements in the analysis results of the Unmix model, Factor 1 can be identified as the impact of the transportation system, Factor 2 is the impact of natural and manmade mixed factors, and Factor 3 is the influence of natural factors. The distribution of each element among Factors and sources is consistent, indicating that the division of source types is credible.

**Table 7.** Principal component analysis of heavy metal contents in dust.

| Element | Cr | Mn | Ni | Cu | Zn | As | Pb | V | Co |
|---|---|---|---|---|---|---|---|---|---|
| Factor 1 | −0.513 | 0.412 | −0.578 | 0.245 | 0.784 | 0.686 | 0.778 | −0.012 | −0.673 |
| Factor 2 | 0.633 | 0.391 | 0.734 | 0.198 | 0.45 | 0.535 | 0.366 | 0.204 | 0.688 |
| Factor 3 | −0.351 | 0.706 | −0.067 | −0.516 | −0.185 | −0.095 | −0.104 | 0.91 | 0.121 |
| Cumulative contribution rate% | 32.995 | 58.205 | 78.088 | 87.49 | 91.804 | 95.711 | 97.886 | 99.349 | 100 |

In the principal component analysis results for Cu, Factor 1 and Factor 2 have significant effects on Cu, and the eigenvalues of Factor 1 and Factor 2 differ by 0.047, which is at a low level. Therefore, it is believed that the accumulation of Cu comes from transportation system factors and mixed factors. The Unmix model analysis results show that Cu comes from natural sources and mixed sources. In view of the diversity of Cu sources, the analysis results are considered credible, and the principal component analysis results can be used as supplements. The sources of Cu include transportation systems, natural sources, and mixed sources.

Regarding our final research goal, if we want to effectively prevent heavy metal pollution, we need to clearly identify the sources of heavy metals. At present, with regard to the composition of mixed sources, except for soil parent materials and traffic factors, the impact of other human activities is not clear. Land use type, as an important form of human activity, has a significant impact on the accumulation of heavy metals. The study area in this paper is the main urban area of Tianshui City. The land use in the study area is mainly for transportation, residential land, cultural and educational land, park green space, commercial land, and industrial land [14,17,22,24]. To meet the needs of people's daily activities, the types and functions of land use are divided, the functions are diverse, and the boundaries are unknown. At the same time, using planar land-use patterns to reveal the sources of heavy metal pollution at individual sampling points, results in a certain degree of uncertainty.

Therefore, this paper collects POI data in the study area, replaces land use types with POI functional attributes, and uses elements (Cr, Ni, Cu, Co) mainly from mixed sources as research cases. A buffer with a radius of 160 m (the minimum distance between sampling points) is created around the sampling points, POI samples are extracted in the buffer, and we analyze the correlation between the number of POIs of different functions and the pollution index, exploring the impact of land use. The correlation analysis results in SPSS show that the correlation between the Ni and Co contents and the number of industrial POIs passed was significant at the 0.01 and 0.05 levels, although the correlation coefficients were 0.38 and 0.34, respectively, i.e., less than 0.6. Moreover, Cr and Cu failed the significance test, and there was no correlation. Therefore, the type of land use has little effect on the existence of heavy metals from mixed sources, and it appears as a single weak effect on industrial land. This may be another manifestation of the small explanatory power (0.083 and 0.0396) of the distance from the factory (X3) to Ni and Co in the geographic detection results.

## 5. Conclusions

First, the accumulation characteristics and health risks of heavy metals in the main urban areas of Tianshui City were evaluated by the geoaccumulation index and health risk index, and heavy metals were determined. Then, GeoDetector, the Unmix model, and principal component analysis were used to comprehensively identify the sources of heavy metals. The research results show that there are significant differences in the spatial distribution of different heavy metal pollution levels in Qinzhou District and Maiji District of Tianshui City and that there are also large differences in the pollution levels of different heavy metals in the same area, with serious Cu, Zn, and Pb pollution. The exposure to non-carcinogenic heavy metals in the study area is greater than that to carcinogenic heavy metals. The exposure to Cr is the largest. The total exposure of children to heavy metals in dust is much greater than that of adults, which is consistent with the conclusions of other researchers [61,62]. The exposure of adults to carcinogenic heavy metals is more significant than that of children. The exposure effects of non-carcinogenic heavy metals to different groups are different. The digestive system is the main pathway of exposure and has the highest non-carcinogenic risk quotient. Dust and heavy metals have a strong non-carcinogenic risk to children and adults; children have a more significant non-carcinogenic risk than carcinogenic risk, and they are more sensitive to the health risk of Pb pollution. Under the same pollution conditions, children's hand-to-mouth frequency (playing and

unconscious behaviors) and respiration rate per unit weight are higher than adults [63,64]; thus, children face greater health risks, and special attention should be given to their health. Furthermore, in view of children's sensitivity to Pb pollution, when severe Pb accumulation occurs in cities, it is recommended to test Pb levels in children's blood.

The traceability analysis results show that the soil parent material factor has the strongest explanatory power for the accumulation of heavy metal elements, followed by the transportation system factor. Industrial and other social factors have explanatory power but are at a low level. Zn, Pb, and As mainly come from the transportation system; V comes from the soil-forming parent material; and Mn, Ni, Cu, and Co come from a mixed source composed of anthropogenic and natural sources. The anthropogenic source is mainly composed of population activity intensity, urban construction, urban economic activities, and industrial production, reflecting the impact of a single industrial system. Compared with cities where heavy metal pollution originates primarily from industrial and agricultural production [17,22,47,61,62,65], the heavy metal pollution in the main urban area of Tianshui City arises mainly from the transportation system. For this type of city, in which urban traffic is the main source of heavy metal pollution, more efforts may be needed to optimize the layout and power of the urban transportation system, promote clean energy vehicles, reduce vehicle emissions and brake wear, and control the accumulation of heavy metal elements.

In addition, some key sources of heavy metals may vary in time. Therefore, it is necessary to continue to explore the time-varying characteristics of heavy metal pollution and sources to adopt more targeted prevention and control strategies to reduce the risk of heavy metals in dust [66].

**Author Contributions:** Conceptualization, B.T. and H.W.; methodology, B.T. and X.W.; software, C.M.; investigation, B.T. and X.W.; visualization, B.T. and J.Z.; writing—review and editing, B.T.; supervision, X.D.; funding acquisition, H.W. and X.W. All authors have read and agreed to the published version of the manuscript.

**Funding:** This work was supported by the National Natural Science Foundation of China (grant numbers 41861037).

**Data Availability Statement:** Third party data, not applicable.

**Acknowledgments:** We thank the editor and the reviewers for their insightful and valuable suggestions, which greatly improved the quality of this manuscript.

**Conflicts of Interest:** The authors declare no conflict of interest.

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
