# Peer review of "Health Risks and Source Analysis of Heavy Metal Pollution from Dust in Tianshui, China"

_minerals, doi:10.3390/min11050502_

Round 1

Reviewer 1 Report

The presentation is quite clear - congratulations. I have never been in this area. Some questions remain:

The calculation formula of the geo-accumulation index is wrong. The tolerance factor of 1,5 has to be multiplied with the background concentration. Maybe this changes fig. 2

formulas after line 153: replace Lngr by IngR   ... ingestion rate

Table 2) You have assumed an exposure frequency for persons of 6 and 24 years of age, of 180 days within a year for non-cancerogenic risk , - like in Luanda ... why? Luanda is a big harbour city at the African West coast, and your site is different! The cancerogenic risk has been calculated for 70 years.

Lines 232-235: delete insignificant commas - this is not as exact as mathematics assume

How to conclude from deposited dust to inhaled dust? You have data from deposited dust < 83 µm from 51 sampling sites, but no concentrations in air. Because the residence time of smaller particles in air is much larger, and thus transport from sources, this causes some uncertainty .

Coarser particles are depleted in the nose and the throat, and the finest exhaled again; the intermediates remain in the lung. Thus, your risk assessment from total deposited dust is just a very rough estimation.

In case of chromium, worst case has been assumed, that means, all Cr is heaxavalent. The authors should mention this. Trivalent Cr is not toxic so far. Speciation measurement by XRF is not possible, however. Cr(VI) in dust may originate from cement killns and smelters (basic slag), that is component X10 in the text. Cr from corrosion of cars and erosion of cars (component X1), however, is likely to be trivalent, because of reduction with humics, washout of hexavalent and the like. The high risk of Cr, is therefore, unlikely.

Figure 3: I would prefer to place the graph of Co between Mn and Ni, Its natural abundance is similar to Ni, but it has also a high affinity to MnOx. Ni and V have crude oil as a common source, but this depends on the source of oil, and may not be true for the investigated site.

Line 390: in my country, sources of Zn are also roofs and gutters, in particular washout thereof - the authors may check this.

Author Response

Please check Response to Reviewer 1.

Reviewer 2 Report

This article has interesting data, and that data is explored with good tools to explain it and draw conclusions.

A small suggestion is to bold format significant correlations in table 8 for factors 1 to 3. This is explained in lines 461-463 but the bold format of those 9 values on that table helps the reader.

figures 3 and figure 5 contains interpolation surfaces from point data that have strange features/smoothness. Please check if there are erroneous "filters" or other parameters "influence radius", etc. on kriging interpolation. I do not have the point data to try it or theoretical expertise to help/comment further on this!

There are small and few mistakes in English that any "pseudo-automatic word processor checking" can help to clean.

Above all, the paper has scientific soundness.

Author Response

Please check Response to Reviewer 2.

Reviewer 3 Report

Dear Authors,

Generally, the manuscript (MS) fits into the scope of the Sustainability journal. The structure of MS respects Scientific Best Practice. In my point of view, the MS needs in some transformation. Here is a list of corrections to be made to the text.

1) Laboratory Determination (p. 2.2). Please, add information about Certified reference material using during analitical procedure and the recoveries for all elements  It is suggested to include detection limit (DL) of all elements especially those below DL. 

2) Discussion (p. 4). If the authors want to conclude generally, then they have to fall trace elements into two main categories: geogenic or those affected more by anthropogenic activities. I suggest to consider anthropogenic sources of trace element concentrations according to their respected land uses. 

3) In the conclusions, in addition to summarising the actions taken and results, it is recommended to use quantitative reasoning comparing with results of previous work.

4) Please, improve the readability of Figures 3 and 5.

Author Response

Please refer to Response to Reviewer 3.
